# Typed Formulation of Classicality

Sanad Kadu, Nicholas LaRacuente, Amr Sabry

Indiana University

Over the years, people have proposed effectful extensions [2–4,7] to $\Pi$, the internal language of rig groupoids [8], to construct approximately universal languages for unitary (pure) quantum computation. These metalanguages provide combinator semantics for implementations of reversible languages like ISO [15] (based on [9]), Qunity [9, 13], which enforce quantum control by allowing the superposition of terms and unitary operations. The principle of deferred measurement is implicitly used to obtain the classical result at the end of the program execution while maintaining reversible foundations. Heunen and Kaarsgaard [7] extend $\Pi$ using universal categorical constructions to model classical cloning and hiding as arrow constructions [1] over base language, thus adding measurement as an irreversible computational effect in an otherwise reversible language. The resulting language though universal for quantum channels, supports only quantum data types and provides no distinction between the type of a classical bit and a qubit, leading to unintuitive type signatures for the measurement operation, i.e., **Qbit** $\rightarrow$ **Qbit** instead of **Qbit** $\rightarrow$ **Bit**. We build upon their work and fix this issue by formulating a hybrid classical/quantum arrow metalanguage using the **Split**$^{\overset{\perp}{=}}$ construction [6, 10], adding classical types as a computational effect to $U\Pi_a^\chi$ [7] by taking its semantics from the rig category of Hilbert spaces and CPTP maps to the category of finite-dimensional $C^\star$-algebras and partial quantum channels. We present steps toward a distinction between classical bits and quantum qubits, allowing the expression of quantum channels as programs in the metalanguage. The equality of such programs/channels can only be specified approximately, as the maps involved come from a continuous set. We study this notion of approximate equality of channels and build towards a fault-tolerant equational theory. This work aligns with recent efforts to study quantum information and computation from a categorical point of view.

To use facts specific to quantum theory using $\Pi$ the monoidal sum of tensor unit $I + I$ is considered a type of qubit instead of a bit. This allows the interpretation of combinators in the language to live in categories specific to quantum theory. The language can then be extended using arrow constructions which semantically correspond to categorical completions. For example, each succeeding arrow metalanguage in the three generations of languages $U\Pi, U\Pi_a, U\Pi_a^\chi$ presented in [7] have their interpretations in the preceding category of the language over which arrows are constructed. When measurement is defined abstractly as an arrow over $U\Pi_a$ which has its semantics in $R[\textbf{Unitary}]$ it does avoid the functional-analytic semantics using operator algebras [5, 11, 12, 14] but introduces the aforementioned problem of bits vs qubits.

An idempotent on an object $A$ of some category is a morphism $p : A \rightarrow A$ such that $p \circ p = p$. Such a $p$ is said to be a *splitting idempotent* when it can be decomposed by a pair of morphisms $(m, e)$ where $m : B \rightarrow A$ and $e : A \rightarrow B$ such that $p = m \circ e$. If a category admits discarding i.e. there is an irreversible unique map $\overset{\perp}{=}_A : A \rightarrow I$ for all objects (e.g. the discard map of $U\Pi_a^\chi$ which is interpreted in $\textbf{L}[\textbf{R}[\textbf{Unitary}]]$) then, an idempotent in such a category is said to *split casually* when it has a splitting $(m, e)$ with $m$ *casual* (i.e., $\overset{\perp}{=}_A \circ m = \overset{\perp}{=}_B$).

Such splitting can be freely added to any category $C$ using the *Karoubi envelope* construction, denoted **Split**$(C)$, where: objects are pairs $(A, p)$ with $p : A \rightarrow A$ an idempotent and morphisms $f : (A, p) \rightarrow (B, q)$ are morphisms $f : A \rightarrow B$ in $C$ satisfying $f = q \circ f \circ p$. The identity on $(A, p)$ is given by $p$. Following Remark 2.8 of [6], the **Split** construction can be seen as a way to introduce new objects by imposing a fundamental restriction on the allowed morphisms—namely, those that satisfy $f = q \circ f \circ p$. We argue that these new objects can be used to interpret classical types. The idea, then, is to select a specific class of idempotents in the category interpreting $U\Pi_a^\chi$ that have a natural splitting, and construct an arrow metalanguage that can be interpreted in **Split**$^{\overset{\perp}{=}}(\textbf{L}[\textbf{R}[\textbf{Unitary}]])$. This captures the inherent decoherence structure of the measure combinator of $U\Pi_a^\chi$ in a way that is compatible with the monoidal structure of the existing category.

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
