# OpenReview forum: "Typed Formulation of Classicality"
_purdue.edu/Purdue_University/PQAI/2025/Symposium — PQAI 2025 Poster_

### Official Review · Reviewer_hw8B · 2025-07-25
**From bits to qubits**

**Rating:** 6
**Confidence:** 2

**Review:**

This abstract addresses a relevant issue in quantum programming languages: the lack of a clear type distinction between classical and quantum data (e.g., bit vs. qubit). The authors build on existing theoretical work and propose a refined type system using a categorical construction to model classical data more precisely. The technical details are quite abstract, but the motivation is clear and the direction is meaningful. As a presentation, the contribution is appropriate and valuable, and I recommend accept.

---

### Official Review · Reviewer_YCAp · 2025-07-25
**To be improved**

**Rating:** 5
**Confidence:** 2

**Review:**

The contribution “Typed Formulation of Classicality” by Kadu et al. builds upon previous works of the same authors (or a subset of them) and is about adding measurement operations into languages for quantum computations.
I am not an expert in the topic of the submission. The content seems to me coherent, but I can’t judge deeply on its quality. However, I have a few general criticisms:
- the submission is only one page long: this is very short, even for a poster submission;
- the content seems to me oriented to Quantum Logic: as such, it would be outside the range of topics covered by PQAI;
- the references are not well structured; they lack important data (year of publication, journal, …) which are necessary to univocally identify the papers in the bibliography.
If the paper gets accepted as a poster contribution, based on my recommendation and on the final decision, I urge the authors to improve their references to an acceptable standard for a scientific publication.

---

### Decision · Program_Chairs · 2025-07-29

Accept (Poster)